# Fluorescent Molecularly Imprinted Polymer Layers against Sialic Acid on Silica-Coated Polystyrene Cores—Assessment of the Binding Behavior to Cancer Cells

**DOI:** 10.3390/cancers14081875

**Published:** 2022-04-08

**Authors:** Sarah Beyer, Martha Kimani, Yuecheng Zhang, Alejandra Verhassel, Louise Sternbæk, Tianyan Wang, Jenny L. Persson, Pirkko Härkönen, Emil Johansson, Remi Caraballo, Mikael Elofsson, Kornelia Gawlitza, Knut Rurack, Lars Ohlsson, Zahra El-Schich, Anette Gjörloff Wingren, Maria M. Stollenwerk

**Affiliations:** 1Department of Biomedical Sciences, Faculty of Health and Society, Malmö University, SE-205 06 Malmö, Sweden; sarah.beyer@med.lu.se (S.B.); yuecheng.zhang@mau.se (Y.Z.); louise.sternbaek@mau.se (L.S.); jenny.persson@mau.se (J.L.P.); lars.ohlsson@mau.se (L.O.); zahra.el-schich@mau.se (Z.E.-S.); maria.stollenwerk@mau.se (M.M.S.); 2Chemical and Optical Sensing Division, Bundesanstalt für Materialforschung und -prüfung (BAM), Richard-Willstätter Straße 11, 12489 Berlin, Germany; martha-wamaitha.kimani@bam.de (M.K.); kornelia.gawlitza@bam.de (K.G.); knut.rurack@bam.de (K.R.); 3Biofilms-Research Center for Biointerfaces, Malmö University, SE-205 06 Malmö, Sweden; 4Institute of Biomedicine, University of Turku, 20520 Turku, Finland; alejandra.verhassel@utu.fi (A.V.); harkonen@utu.fi (P.H.); 5FICAN West Cancer Centre, Turku University Hospital, 20520 Turku, Finland; 6Phase Holographic Imaging AB, SE-223 63 Lund, Sweden; 7Department of Molecular Biology, Umeå University, SE-901 87 Umeå, Sweden; tianyan.wang@umu.se; 8Department of Chemistry, Umeå University, SE-901 87 Umeå, Sweden; emil.o.johansson@ilk.uu.se (E.J.); remi.caraballo@scilifelab.se (R.C.); mikael.elofsson@umu.se (M.E.); 9Umeå Centre for Microbial Research, Umeå University, SE-901 87 Umeå, Sweden

**Keywords:** cancer, imprinting, molecularly imprinted polymers, SA conjugates, sialic acid

## Abstract

**Simple Summary:**

Cancer cells often have aberrant sialic acid expression. We used molecularly imprinted polymers in this study as novel tools for analyzing sialic acid expression as a biomarker on cancer cells. The sialic acid imprinted polymer shell was synthesized on a polystyrene core, providing low-density support for improving the suspension stability and scattering properties of the molecularly imprinted particles compared to previous core-shell formats. Our results show that these particles have an increased ability to bind to cancer cells. The binding of these particles may be inhibited by two different pentavalent sialic acid conjugates, pointing to the specificity of the sialic acid imprinted particles.

**Abstract:**

Sialic acid (SA) is a monosaccharide usually linked to the terminus of glycan chains on the cell surface. It plays a crucial role in many biological processes, and hypersialylation is a common feature in cancer. Lectins are widely used to analyze the cell surface expression of SA. However, these protein molecules are usually expensive and easily denatured, which calls for the development of alternative glycan-specific receptors and cell imaging technologies. In this study, SA-imprinted fluorescent core-shell molecularly imprinted polymer particles (SA-MIPs) were employed to recognize SA on the cell surface of cancer cell lines. The SA-MIPs improved suspensibility and scattering properties compared with previously used core-shell SA-MIPs. Although SA-imprinting was performed using SA without preference for the α2,3- and α2,6-SA forms, we screened the cancer cell lines analyzed using the lectins Maackia Amurensis Lectin I (MAL I, α2,3-SA) and Sambucus Nigra Lectin (SNA, α2,6-SA). Our results show that the selected cancer cell lines in this study presented a varied binding behavior with the SA-MIPs. The binding pattern of the lectins was also demonstrated. Moreover, two different pentavalent SA conjugates were used to inhibit the binding of the SA-MIPs to breast, skin, and lung cancer cell lines, demonstrating the specificity of the SA-MIPs in both flow cytometry and confocal fluorescence microscopy. We concluded that the synthesized SA-MIPs might be a powerful future tool in the diagnostic analysis of various cancer cells.

## 1. Introduction

Sialic acid (SA) is a nine-carbon monosaccharide located at the terminal end of cell surface proteins, lipids, or secreted proteins [1]. SA can be linked to the C-6 position of N-acetylgalactosamine (GalNAc), to the C-6 or C-3 positions of galactose (Gal), or the C-8 or C-9 positions of another SA [2]. SA plays a critical role in many normal physiological and pathological processes, e.g., the repulse of cell–cell interaction, and serves as a binding site for different toxins, pathogens, and glycan-binding proteins [1]. The cellular change in response under physiological condition often results in the dynamic regulation of the cell surface glycosylation pattern [3]. The hypersialylation accelerates cancer progression and can lead to a poor prognosis [4]. The special metabolic flux and aberrant expression of sialyltransferases/sialidases mainly cause increased SA expression in tumor cells.

Moreover, the level of SA expression in cancer has been shown to result in the cancer cell’s increased metastatic and invasive potential [5]. Traditionally, SA expression has been analyzed by using lectins such as Maackia Amurensis Lectin I (MAL I), Sambucus Nigra Lectin (SNA), and by antibodies targeting SA [6]. However, specific targeting of glycosylated proteins is challenging since the availability of adequate lectins and glycan-specific antibodies is limited [7]. The specificity and affinity of the lectins and antibodies are usually not sufficient, which calls for the development of alternative glycan-specific receptors and cell imaging technologies.

Boronic-acid-based semi-covalent imprinting is widely used to recognize glycoproteins since they bind reversibly with cis-diol groups of the saccharide units [8]. Molecularly imprinted polymers (MIPs) are polymers that incorporate a target template in the polymerization process. Template removal after synthesis leaves cavities in the polymer matrix with high affinity and specificity to the target molecule [9,10]. Being of non-biological origin, engineered MIPs are extremely robust, resist denaturing solvents, are stable at high temperatures, and can be reproduced at a low cost [11]. Indeed, MIPs have been successfully applied as artificial recognition elements in targeting glycans such as SA and glycosaminoglycans (GAGs) [10,12,13,14,15,16]. We, and others, used monosaccharide SA as a template and developed core-shell SA-imprinted particles with a defined size, which can be further applied as imaging agents for cell surface glycans [17,18,19,20,21,22]. We reported the use of a nitrobenzoxadiazole (NBD) fluorochrome as a reporter group and SA as an imprinted template on the surface of silica beads of approximately 200 nm [10,17,22,23,24,25]. Kimani et al. showed that non-imprinted particles (NIPs) prepared with a “dummy” template served as a better negative control in cell-binding assays than the common template-free control NIPs [22].

We recently reported these particles’ synthesis and binding properties in solution and cell labeling assays with two cancer cell lines, the epidermal carcinoma cell line A431 and pulmonary epithelial carcinoma cell line A549 [22]. In this study, we used SA-MIPs synthesized on silica-coated polystyrene (PS) particles (ca. 170 nm) as cell staining agents. Compared to previous works, where silica beads have been used as the core [17], a polystyrene core provides lighter lower-density support for improved suspensibility and scattering properties [26]. We analyzed the SA expression in thirteen different cancer cell lines by flow cytometry and confocal fluorescence microscopy using SA-MIPs. Our results showed that the different cancer cell lines show varied expressions of α2,3- and α2,6-SA, thus displaying different binding behaviors to SA-MIPs.

In addition, the specificity of the SA-MIPs was validated using two pentavalent SA conjugates, ME0752 and ME0976 [27]. By pre-treating the SA-MIPs with different concentrations of the two SA conjugates prior to cell staining using both flow cytometry and confocal fluorescence microscopy, we observed a reduction in SA-MIP binding to the cells. Our results showed the potential of applying SA-MIPs to test complex biological samples. We concluded that synthesized SA-MIPs could be a powerful tool in the diagnostic analysis of cancer cells.

## 2. Materials and Methods

### 2.1. Reagents and Cell Culture

The biotin-labeled lectins MAL I and SNA were purchased from Vector Laboratories (Burlingame, CA, USA). The streptavidin-fluorescein isothiocyanate (FITC) was obtained from Agilent Technologies (Santa Clara, CA, USA). The Falcon multi-chamber culture glass slides were purchased from Corning (Glendale, AZ, USA). The mounting medium ProlongQR Gold antifade reagent, phosphate buffered saline (PBS), vinylbenzene boronic acid (VBBA), and 4′,6-diamino-2-phenylindole (DAPI) were bought from Thermo Fisher Scientific (Waltham, MA, USA). Triton 100X, methacrylamide (MAAm), ethylene glycol dimethacrylate (EGDMA), formaldehyde, and rhodamine phalloidin were purchased from Sigma–Aldrich (Taufkirchen, Germany). The 2,2′-Azobis(2,4-dimethylvaleronitril) (ABDV) initiator was obtained from Wako Chemicals (Neuss, Germany). Human cell lines including MDA-MB-468, MDA-MB-231, CAMA-1, T-47D, MCF7, SK-BR-3, Hs 578T, A549, A-431, PC-3, THP-1, and Jurkat were obtained from American Type Culture Collection (ATCC) (Manassas, VA, USA). Hek-n cells are primary human epidermal keratinocytes isolated from neonatal foreskin and were purchased from Thermo Fisher Scientific (Waltham, MA, USA). The following cells were cultured in a cell culture medium purchased from Thermo Fisher Scientific (Waltham, MA, USA): MDA-MB-231 and MDA-MB-468 cells were cultured in Dulbecco’s Modified Eagle Medium (DMEM) supplemented with 10% fetal bovine serum (FBS). MCF7, Jurkat, THP-1, and T-47D cells were cultured in Roswell Park Memorial Institute (RPMI) 1640 medium supplemented with 10% FBS and 50 µg/mL gentamycin. Hs-578T cells were cultured in DMEM supplemented with 10% FBS, 1% penicillin-streptomycin (PEST), and 10 µg/mL insulin. CAMA-1 was cultured in RPMI-1640 medium supplemented with 10% FBS, 1% PEST, and 1% sodium pyruvate. A549 cells were cultured in RPMI-1640 medium supplemented with 10% FBS and 1% PEST. SK-BR-3 and PC-3 were cultured in DMEM supplemented with 10% FBS, 1% GlutaMAX (Thermo Fisher Scientific, Waltham, MA, USA), and 1% PEST. Hek-n cells were maintained in EpiLife growth medium with 60 mM calcium chloride supplemented with 1% human keratinocyte growth supplement (HKGS) and 0.2% gentamycin/amphotericin. A-431 cells were cultured in Eagle’s Minimum Essential Medium (EMEM, Sigma-Aldrich, St. Louis, MO, USA) supplemented with 10% FBS, 1% L-glutamine and 1% non-essential amino acids (Thermo Fisher Scientific, Waltham, MA, USA). All the cell lines were cultured at 37 °C with 5% CO_2_ in 100% humidity. 

### 2.2. SA-MIP Synthesis

The synthesis of SA-MIPs was performed as we recently reported [22]. In brief, 0.9 mg MAAm, 1.8 mg 2-(3-(4-nitrobenzo[c][1,2,5]oxadiazol-7-yl)ureido) ethyl methacrylate, 0.8 mg VBBA, and 41 µL EGDMA were dissolved in 2 mL *N,N*-dimethylformamide (DMF) and sonication was performed for 15 min. Afterwards, the pre-polymerization mixture was mixed with 20 mg RAFT@SiO_2_@PS particles and 1.8 mg of the SA template (dissolved in 500 µL DMF). This mixture was sonicated for 20 min followed by a 20 min incubation in a heater at 50 °C at 500 rpm with a degassing process. 600 µL ABDV in DMF solution (4.5 mM) was added to the vial, and the reaction continued for 22 h. After incubation, the particles were washed several times to remove the template. In the final step, the MIP particles were dried overnight at room temperature under vacuum.

### 2.3. Flow Cytometry Assay for Lectin Staining

Briefly, the cells were first washed twice with 2 mL PBS. Next, the cells were divided into several flow cytometry tubes (5 × 10^5^ cells per sample), and a 100 µL mixture of cells, 5 µg/mL lectins, and PBS were incubated in the dark at 4 °C for 30 min. These samples were washed twice with 1.5 mL PBS followed by staining with 10 µg/mL of streptavidin-FITC in the dark at 4 °C for 20 min. After incubation, the cells were again washed twice and resuspended in 300 μL PBS for flow cytometric analysis. The flow cytometric analysis was conducted on an Accuri C6 flow cytometer (BD Biosciences, Franklin Lakes, NJ, USA) with a 488 nm excitation laser coupled to a 530/30 nm bandpass (BP) filter. Ten thousand events were captured in the gate and used for the SA-expression analysis.

### 2.4. Flow Cytometry Assay for MIPs Staining

A total of 1 × 10^6^ cells per sample were stained with SA-MIPs. The cells were washed twice with 2 mL PBS and divided thereafter into several flow cytometry tubes and incubated with SA-MIPs (0.1 mg/mL) at 4 °C for 30 min in the dark. After incubation, the cells were washed twice and resuspended in 300 μL PBS for flow cytometric analysis. Ten thousand events were captured in the gate and used for the SA-expression analysis.

### 2.5. Pre-Treatment of SA-MIPs with Pentavalent SA Conjugates

In this assay, the SA-MIPs were pre-treated with pentavalent SA conjugates (SA conjugates) before being used in a MIP staining assay according to the experimental procedure described above. Pre-treatment was conducted by pre-incubating the MIP in PBS with different concentrations (20 and 200 µM) of the SA conjugates ME0976 [27] or ME0752 [27] at room temperature for 5 min.

### 2.6. Confocal Fluorescence Microscopy Analysis

For the confocal fluorescence microscopy analysis, 1 × 10^5^ cells per well were seeded in a Falcon multi-chamber culture glass slide with a final volume of 500 μL and incubated for 48 h at 37 °C with 5% CO_2_ at 100% humidity. The cells were washed in PBS and fixed with 100 μL 4% formaldehyde at room temperature for 10 min followed by washing twice with PBS and once with 0.05% Triton 100X in PBS. Afterwards, the cells were permeabilized and stained with 100 μL of 1/100 diluted rhodamine-phalloidin for 30 min in the dark at room temperature. After washing twice with 0.05% Triton 100X in PBS and twice with PBS, the samples were incubated with 100 μL SA-MIPs (0.1 mg/mL), either untreated or pre-treated as described above, with 200 µM SA conjugates for 30 min in the dark at room temperature. The samples were washed four times with PBS and stained with 300 nM DAPI in the dark for 4 min at room temperature. After another two washes with PBS, the samples were mounted with one drop of mounting medium ProlongQR Gold antifade reagent and stored at 4 °C before analysis by confocal fluorescence microscopy (Nikon Instruments A1R HD25, Melville, NY, USA).

## 3. Results

### 3.1. SA Staining with Lectins MAL I and SNA of a Variety of Cancer Cell Lines

In this study, two different lectins, MAL I (α2,3-SA) and SNA (α2,6-SA) were used to analyze the expression of SA on the surface of thirteen human cancer cell lines by flow cytometry: SK-BR-3, MDA-MB-468, PC-3, THP-1, Jurkat, A-431, MCF7, MDA-MB-231, A549, Hek-n, CAMA-1, T-47D, and Hs 578T cells. The α2,3-SA and α2,6-SA lectin staining results are presented in histograms (Figure 1). The MFI of unstained control cells as the background is also shown for each cell line in Figure 1 (black line). The breast cancer cell line CAMA-1 shows the lowest expression of α2,3-SA. For α2,6-SA expression, the breast cancer cell line MCF7 and prostate cell line PC-3 show the least pronounced expression. High expression levels of α2,3-SA were determined in the breast cancer cell lines Hs 578T and MDA-MB-231 and the lung carcinoma cell line A549. The experiment has been repeated twice with minor deviations. Therefore, one representative experiment out of two performed is shown. 

### 3.2. The Binding Patterns of SA-MIPs to Several Cancer Cell Lines

The binding properties of the SA-MIPs were analyzed using flow cytometry. The MFI of SA-MIP binding to the cell lines SK-BR-3, MDA-MB-468, PC-3, THP-1, Jurkat, A-431, MCF7, MDA-MB-231, A549, Hek-n, CAMA-1, T-47D, and Hs 578T are shown in Figure 2. The SA-MIP binding properties are displayed in histograms (Figure 2A) and bar diagrams showing the MFI (Figure 2B). The order of the cell lines is based on the binding capacity of the SA-MIPs.

### 3.3. Pre-Treatment of SA-MIPs with SA Conjugates to Reduce Binding to Cancer Cell Lines

Three cell lines with different binding properties to SA-MIPs, A549 lung carcinoma cells (high binding), MCF7 breast cell line (average binding), and A-431 skin carcinoma cells (low binding), were selected for further analysis. The specificity of the SA-MIP binding was assessed using flow cytometry. The SA-MIPs were pre-incubated with SA-conjugates ME0752 or ME0976 at a concentration of either 20 µM or 200 µM, respectively, and then applied to cell-binding assays. For all three cell lines, a reduction in MFI was observed for SA-MIP binding to cells after incubation with both concentrations of SA-conjugates (Figure 3A–C). The highest reduction occurred in the skin carcinoma cell line A-431 (Figure 3B).

### 3.4. SA-MIPs Staining Patterns Visualized by Confocal Fluorescence Microscopy

To visualize the binding of the SA-MIPs to the cells and characterize the fluorescence properties of the SA-MIPs, the three selected cancer cell lines, A549, MCF7, and A-431 were analyzed using confocal fluorescence microscopy. In addition to staining with SA-MIPs, all cell lines were stained with DAPI and phalloidin for nuclei and cytoskeleton visualization, respectively. The binding pattern and distribution of the SA-MIPs were different in the three cell lines (Figure 4A–L).

The A549 cells showed a uniform distribution of SA-MIPs (Figure 4B). In contrast, the A-431 cells showed a very low binding of the SA-MIPs (Figure 4F), whereas MCF7 cells showed a heterogenous SA-MIP binding pattern (Figure 4J).

The SA-MIP binding pattern changed after pre-treatment with 200 µM of SA conjugate ME0752. The A549 cells showed less binding of SA-MIPs (Figure 4C). In contrast, A-431 cells showed that several SA-MIPs were bound as small particles (Figure 4G). MCF7 cells showed similar amounts of bound particles (Figure 4K).

After pre-treatment with 200 µM of SA conjugate ME0976, the SA-MIP binding pattern for A549 cells (Figure 4D) and A-431 cells (Figure 4H) remained similar to pre-treatment with ME0752. The MCF7 cells showed fewer bound particles than those pre-treated using ME0752 (Figure 4L).

## 4. Discussion

We previously reported about developing and using core-shell SA-imprinted particles for determining cell surface glycans [10,17,22,23,24,25]. Here, we extended the cell surface SA analysis by utilizing core-shell SA-MIPs developed on a silica-coated polystyrene core [22]. 

All thirteen cancer cell lines were stained with the improved SA-MIP batch and analyzed by flow cytometry. Our results revealed a different staining pattern for the cancer cell lines, which cannot be compared with the α2,3-SA, and α2,6-SA expression since the SA-MIPs were imprinted with SA. In addition, the weak nature of glycan-mediated interactions may affect the MAL I and SNA binding properties, as well as specificity [7,28].

We further developed the use of pentavalent SA conjugates to validate the binding of the SA-MIPs to three different cell types, A549 lung carcinoma, MCF7 breast cancer, and A-431 skin carcinoma cells [27,29]. The SA conjugates contain flexible spacers (Figure 3D) capable of binding the SA-MIPs in a concentration-dependent manner, and we could show a reduced cell-binding after pre-incubating the SA conjugates with the SA-MIPs. The MFI values were substantially reduced using either conjugates, ME0976 or ME0752 because the SA-specific sites bound the SA conjugates to the MIP particles rendering them unavailable to the SA on the cell surface of the cancer cells. This finding confirms the selectivity of the MIP particles for binding SA. We recently demonstrated the novel use of SA conjugates ME1057 and ME0970 by pre-treating the SA-MIPs (200 nm) with these SA conjugates as inhibitors [30]. It has also been shown that the titration of SA-MIPs with possible competitors GalNAc and glucose did not induce a fluorescence increase [22]. These results, together with our study results, suggest that the SA conjugates represent promising candidates as multivalent inhibitory targets for SA.

The staining patterns of the SA-MIPs were visualized on the selected cancer cell lines, A549, MCF7, and A-431, using confocal fluorescence microscopy. The binding pattern to the lung carcinoma A549 cells revealed a uniform distribution of SA-MIPs. Most interestingly, the addition of pentavalent SA conjugates changed the SA-MIP staining pattern of the cells by diminishing the SA-MIP binding. For all three cell types analyzed, the bound MIPs were more diluted on the cell surface after adding the SA conjugates. A549 cells showed fewer bound particles, whereas MCF7 showed similar numbers. In contrast, the number of SA-MIPs bound to A-431 cells was increased. The preincubation of MIP particles with the SA conjugates prior to cell binding results in the presence of the spacer molecules in the polymer layer of the particles, which facilitates solubilization of the MIP particles in the assay suspension.

In our flow cytometry results, α2,3-SA was expressed at comparable levels in A-431 and MCF7 cells. Confocal microscopy showed very few SA-MIP particles bound to the cell surface of A-431 cells. The different distribution of the SA-MIPs on A549 observed in the confocal microscopy images can be explained by the high expression of both α2,3-SA and α2,6-SA seen in flow cytometry. The MFI values for MAL I staining were high on all cell lines, whereas A549 expressed α2,6-SA to a greater extent, as seen in the SNA staining results.

The SA-MIPs were imprinted without selectivity for the two forms α2,3-SA or α2,6-SA and, therefore, did not reveal clear specificity compared to cell staining using MAL and SNA lectins. Moreover, the SA-MIPs are significantly larger than lectins and can be expected to display multivalent interactions with the cell surface. The fact that each cell line has distinct characteristics and morphology may also influence the binding behavior of the larger MIP particles.

## 5. Conclusions

We analyzed the SA expression in thirteen different cancer cell lines using SA-MIPs together with MAL I and SNA. The results show that the varying expression of α2,3- and α2,6-SA results in different binding capacities for SA-MIPs. Preincubation of the SA-MIPs with pentavalent SA conjugates reduced the overall binding of the MIPs, pointing to the specificity of the MIPs to bind SA. In conclusion, synthesized SA-MIPs may be applied as effective tools to analyze the potential biomarker SA expressed on the surface of cancer cells..

## Figures and Tables

**Figure 1 cancers-14-01875-f001:**
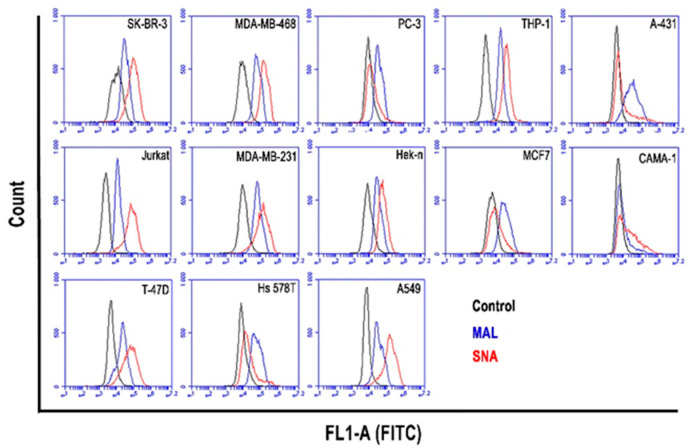
MAL I (α2,3-SA) and SNA (α2,6-SA) lectin binding to the thirteen cancer cell lines. Both lectins were used at a concentration of 5 µg/mL. The flow cytometry histograms show the mean fluorescence intensity (MFI) of unstained control cells (black lines) and lectin-stained cells (blue lines for MAL I and red lines for SNA). One representative experiment out of two performed is shown.

**Figure 2 cancers-14-01875-f002:**
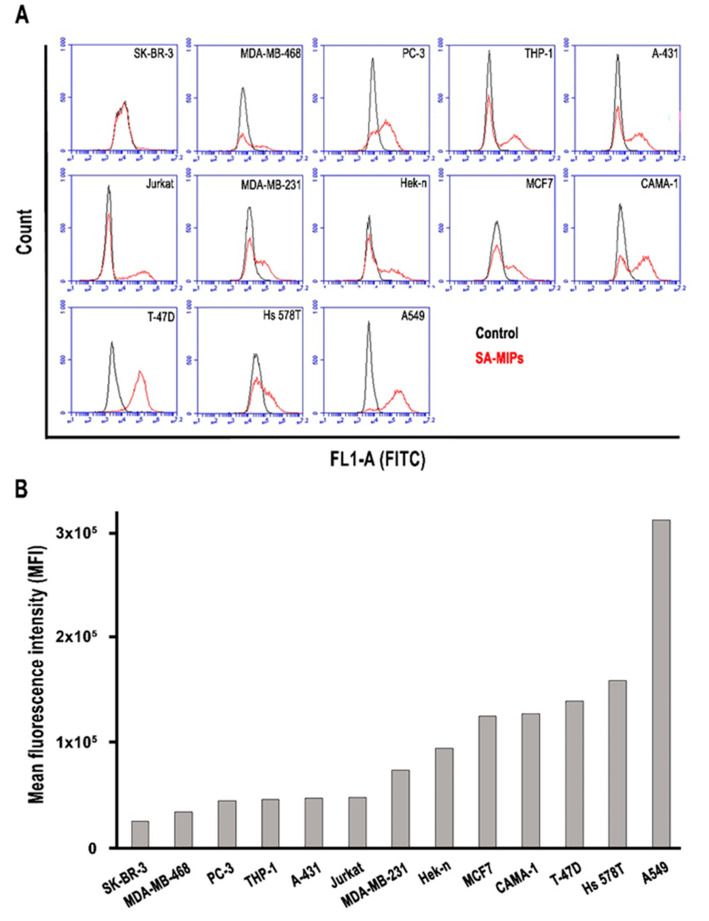
SA-MIP binding to the thirteen cancer cell lines as shown by MFI. The concentration of SA-MIPs is 0.1 mg/mL. (**A**) The flow cytometry histograms show the MFI of unstained control cells (black lines) and SA-MIP-stained cells (red lines); (**B**) the bar diagrams show the MFI of SA-MIP-stained cells. One representative experiment out of two performed is shown.

**Figure 3 cancers-14-01875-f003:**
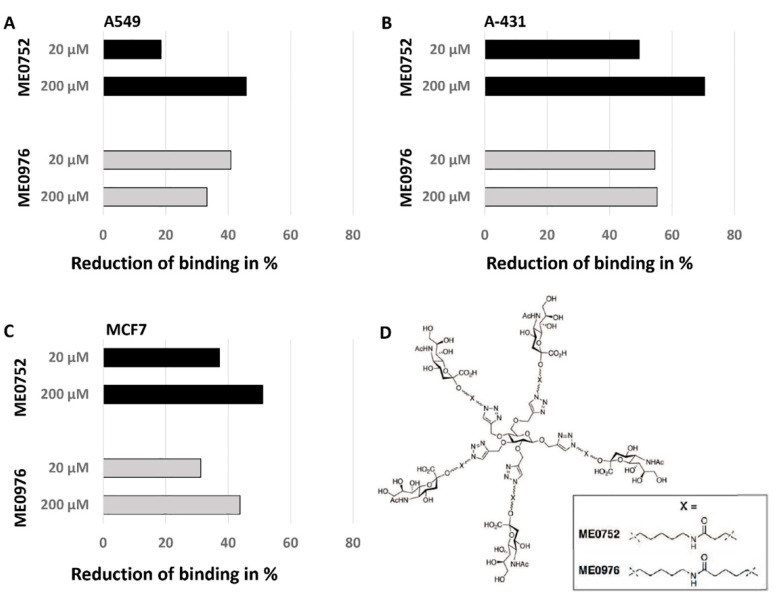
SA-MIPs were pre-incubated with different concentrations of SA conjugates and analyzed with flow cytometry. The reduction in binding compared to SA-MIP binding alone is shown. The SA conjugates ME0752 and ME0976 were added to the SA-MIPs at 20 µM and 200 µM, respectively, and the particles were used thereafter to stain (**A**) A549, (**B**) A-431, and (**C**) MCF7 cells. The chemical structures for ME0752 and ME0976 are shown in (**D**). One representative experiment out of two performed is shown.

**Figure 4 cancers-14-01875-f004:**
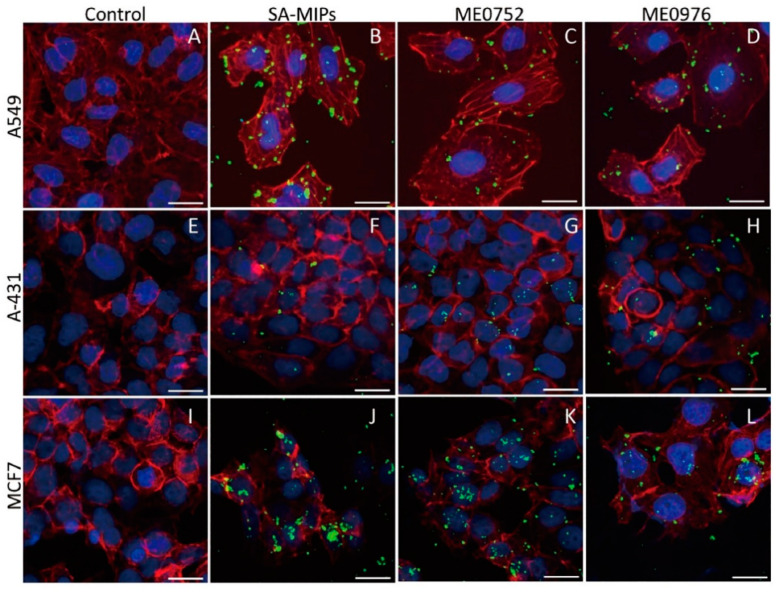
Confocal fluorescence microscopy images of SA-MIPs staining for three different cancer cell lines. A549 (**A**–**D**), A-431 (**E**–**H**), and MCF7 (**I**–**L**) were stained with SA-MIPs (**B**,**F**,**J**) in green, rhodamine-phalloidin (actin filaments) in red and DAPI (nuclei) in blue. The two columns on the right show staining with SA-MIPs after pre-treatment with the 200 µM of SA conjugates ME0752 (**C**,**G**,**K**) and ME0976 (**D**,**H**,**L**). Scale bar: 20 µm.

## Data Availability

The data presented in this study are available in this article.

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
