# Peer review of "Fluorescent Molecularly Imprinted Polymer Layers against Sialic Acid on Silica-Coated Polystyrene Cores—Assessment of the Binding Behavior to Cancer Cells"

_cancers, 2022, doi:10.3390/cancers14081875_

Round 1
Reviewer 1 Report
Although the authors revised the manuscript according to the comments, some issues are still found as the following. Therefore, modifications are still needed to address these issues.
- Using SA-MIP to extend the number of cell lines tested does not seem novel enough, because the binding behaviors of SA-MIP to cancer cells had been analyzed in a previous study (ref. 19).
- The format of the reference has some errors. In ref 16,26,27, the name of the publication should be provided after the name and the title.
- The question 4 of reviewer 1 has not been answered well. In SK-BR-3 cells, the MFI signal of SA-MIP hardly moved as compared with the control groups. (Figure 2) This means SA-MIP can’t target the SA on the cell surface, weather a2,3 or a2,6 linked sialic acid. This conclusion is contradictory to the Figure 1.
Author Response
Although the authors revised the manuscript according to the comments, some issues are still found as the following. Therefore, modifications are still needed to address these issues.
We thank the referee for these suggestions. Please find our replies below.
1. Using SA-MIP to extend the number of cell lines tested does not seem novel enough, because the binding behaviors of SA-MIP to cancer cells had been analyzed in a previous study (ref. 19).
Reply:
We agree that other studies have been performed, both by us and others. However, SA-MIP studies are still quite novel and the biological interactions between MIPs and cells are important to understand before performing studies in vivo. Therefore, we undertook a comprehensive study to understand the MIP-interactions with many different cancer cell types of human origin. Moreover, MIPs used by different research groups may have small differences that could influence the result, making it important to perform the studies on all these cell lines.
2. The format of the reference has some errors. In ref 16,26,27, the name of the publication should be provided after the name and the title.
Reply: the format of the references 16,26,27, are now revised. This will be seen in the revised version of the manuscript.
3. The question 4 of reviewer 1 has not been answered well. In SK-BR-3 cells, the MFI signal of SA-MIP hardly moved as compared with the control groups. (Figure 2) This means SA-MIP can’t target the SA on the cell surface, weather a2,3 or a2,6 linked sialic acid. This conclusion is contradictory to the Figure 1.
Reply:
It is true that the breast cancer cell line SK-BR-3 in our hands bind the SA-MIPs to a very low degree, although the expression of a-2,3 or a-2,6 linked sialic acid, respectively, is shown by ligation with MAL and SNA. As we explained in the revised version of the manuscript, these SA-MIPs were imprinted without selectivity for the two forms α2,3-SA or α2,6-SA, and did therefore not reveal clear specificity in comparison to cell staining using MAL and SNA lectins. Moreover, we have not in this study focused on the different mono-, di-, tri- or even more complex sialic acid forms that exist in nature, and how they are exposed (or not) at the cell membrane surfaces. This will be the next step to hopefully be investigated in our future studies.
Reviewer 2 Report
All Reviewers comments were explained.
Author Response
We thank the referee for previous valuable comments.
This manuscript is a resubmission of an earlier submission. The following is a list of the peer review reports and author responses from that submission.
Round 1
Reviewer 1 Report
Cancer cells often exhibit aberrant expression of sialic acid (SA), which has been considered as a feature of cancer cells. Therefore, the analysis of SA expression on cancer cells is of importance. In this paper, molecularly imprinted polymers (MIP) were used as novel tools to analyze sialic acid expression as a biomarker on cancer cells, showing some strengths of the synthesized antibodies over lectins. However, due to the fact that there are some severe issues in the paper, the reviewers can’t recommend is for publication in Cancers, at least at the present form.
Major issues:
- What is the main contribution and novelty of this work? Many related investigations have been reported in ref. 17, which largely degrades the novelty of this work.
- The authors claim that the varying expression of a2,3 anda2,6-linked sialic acid resulted in the different binding capacities of SA-MIP. It is not a clear conclusion. How did the a2,3 anda2,6-linked sialic acid affect the binding behavior of SA-MIP?
- The authors claim that the SA-MIP used are lighter compared to previous forms of SA-MIP, but there are no data in this manuscript to support this conclusion.
- The author used MAL and SNA to distinguish the a2,3 and a2,6-linked sialic acid on the various cancer cells as the reference, this is acceptable. However, there are some contradictions in SA-MIP groups. In SK-BR-3, the Lectin group shows that this cell overexpressed sialic acid, and the MFI signal of SA-MIP hardly moved as compared with the control groups. This means SA-MIP can’t target the SA on the cell surface. This conclusion is contradictory.
- A varied binding behavior of the SA-MIPs happened when the author used SA-MIP to stain the cancer cells (Figure 3), but the author didn’t demonstrate why the behavior happened. From the manuscript we also can’t conclude that the MIP could distinguish the a2,3 and a2,6-linked sialic acid.
- The authors mentioned that the binding of MIP to SA is specific in the conclusion from the CLSM imaging by preincubating with the SA conjugates. And this conclusion has been reported previously.[1] One may wonder whether or not the authors want to distinguish the a2,3 and a2,6-linked sialic acid by MIP?
Minor issues:
- Line 282-287, the synthesized SA conjugates was used as the inhibitor, and this point has nothing to do with this manuscript, delete this description.
- Quite a few sentences are meaningless or poorly organized, such as, “Kimani et al showed that MIP particles binding SA were more selective compared to nonimprinted polymer (NIP) control particles [17]”, “In this study, we utilized the SA-MIPs synthesized on silica-coated polystyrene (PS) 82 particles (ca. 170 nm) as cell staining agents [17]”, and …
- The reference citation is inadequate, and quite a lot of relevant important reference should be cited, particularly works by B. Sellergren and Z. Liu, such as 1) L. Mavliutova, B. M. Aldeguer, J. Wiklander, B. Sellergren, et al. RSC Advance, 2021, 11, 22409. 2). L. Mavliutova, E. Verduci, S. A. Shine, B. Sellergren, ACS Omega, 2021, 6, 12229; 3) Danyang Yin, Z. Liu, et al. Chemical Communications, 2015, 51, 17696; 4) Shuangshou Wang, Z. Liu, et al. Analytical Chemistry, 2017, 89, 5646; 5) Zikuan Gu, Z. Liu, et al. Angewandte Chemie International Edition, 2021, 60, 2663.
Reviewer 2 Report
In this manuscript the molecularly imprinted polymers properties towards cancer cells binding were described. Imprinted polymeric material is not new - the synthesis and properties were previously described by Authors (ACS Appl. Polym. Mater. 2021, 3 2363). Also a cell binding studies of described polymers were partially described. Therefore, in my opinion, the topic of the manuscript is interesting. Nevertheless, the careful analysis of the manuscript revealed various drawbacks. However, after serious rewriting maintained by complete experimental data, it could be considered for publication in Cancers. Below are major (but not all) problems that could help Authors to improve their manuscript.
- In Materials and Methods section there is no reagents used during synthesis mentioned in Reagents section.
- In SA-MIP synthesis section various abbreviations are used but there is no explanation of these abbreviation in whole text.
- The novelty level is not very high. It is not new material and partially application of SA-MIP in cells binding was described earlier.
- Comparison of SA-MIP binding properties towards other structurally similar to SA analytes - glycans or polysaccharides could be added.
- The quality of Figure 1 and 3A is poor.
I recommend major revision of the manuscript.